# Rethinking Parameter Sharing for LLM Fine-Tuning with Multiple LoRAs

## Abstract

Large language models are often adapted using parameter-efficient techniques such as Low-Rank Adaptation (LoRA), formulated as $y = W_0 x + BAx$, where $W_0$ is the pre-trained parameters and $x$ is the input to the adapted layer. While multi-adapter extensions often employ multiple LoRAs, prior studies suggest that the inner $A$ matrices are highly similar during training and thus suitable for sharing. We revisit this phenomenon and find that this similarity is largely attributable to the identical initialization rather than shared knowledge, with $B$ playing a more critical role in knowledge encoding and transfer. Motivated by these insights, we propose **ALoRA**, an asymmetric multi-LoRA design with multiple $A$ matrices and a single shared $B$ in multi-task fine-tuning, and **Fed-ALoRA**, which shares $B$ across clients in federated fine-tuning under both homogeneous and heterogeneous settings, through a novel matrix decomposition strategy to accommodate heterogeneous ranks across clients. Experiments on commonsense reasoning, math reasoning, multi-task NLP dataset, and federated NLP dataset demonstrate that our methods achieve more balanced performance across tasks with comparable or superior average accuracy relative to existing multi-LoRA approaches.

## 1 Introduction

Large language models (LLMs) have achieved remarkable performance across diverse domains (Achiam et al., 2023; Comanici et al., 2025; Dubey et al., 2024), but the growing scale makes conventional full fine-tuning increasingly expensive. Parameter-efficient fine-tuning (PEFT) addresses this challenge by freezing the pre-trained model and updating only a small subset of parameters, improving efficiency while maintaining performance (Han et al., 2024). Among PEFT methods, Low-Rank Adaptation (LoRA) (Hu et al., 2022) is particularly popular: it decomposes weight updates into trainable low-rank matrices $A$ and $B$, which can be merged into the pre-trained model without extra inference latency.

Recent studies have shown that a single LoRA has limited capacity when handling diverse data distributions (Yang et al., 2024; Cai et al., 2025). A natural extension is to use multiple LoRAs, where each module can specialize in different data modes such as tasks, domains, and distributed clients (Li et al., 2024; Sun et al., 2025; Wu et al., 2024b; Liao et al., 2025). In multi-task fine-tuning, adapters are required to handle task heterogeneity (Liang et al., 2025), and in federated fine-tuning, they should account for client heterogeneity and personalization (Bian et al., 2025). However, naively employing multiple LoRAs also increases computation and communication costs, which makes this approach less efficient.

To address this problem, recent methods explore parameter sharing across LoRA modules to improve parameter efficiency. HydraLoRA (Tian et al., 2024) observes that $A$ matrices trained on different tasks exhibit very high similarity, and proposes a single shared $A$ with multiple $B$s for multi-task fine-tuning. FedSA-LoRA (Guo et al., 2025) reports similar findings in federated fine-tuning and transmits only $A$ matrices for server aggregation with reduced communication costs. These studies attribute the high similarity in $A$ matrices to the shared knowledge.

In this paper, we revisit this similarity phenomenon and find that the similarity of $A$ stems mainly from identical initialization rather than shared knowledge. Our analysis of learning dynamics reveals that $A$ functions largely as a feature projector, while $B$ encodes the domain knowledge. A further exploration shows that sharing $B$ yields more effective knowledge transfer than sharing $A$ in both

multi-task and federated fine-tuning. These insights motivate an interesting but underexplored question: *might sharing the module $B$, rather than $A$, be more effective for parameter and knowledge sharing?* In this paper, we provide a positive answer, with our main contributions as follows.

- We propose **ALoRA**, a new asymmetric multi-LoRA architecture for multi-task fine-tuning. It employs multiple $A$ matrices and a single shared $B$ matrix, where the $A$ matrices are dynamically routed by the inputs. This design enables each $A$ to explore distinct feature subspaces while encouraging knowledge transfer through the shared $B$.

- We propose **Fed-ALoRA**, which communicates only $B$ matrices rather than full LoRA parameters for aggregation on server. It supports both homogeneous and heterogeneous settings with the same and different ranks across clients, whereas existing parameter-sharing federated fine-tuning methods focus only on the homogeneous case. In the homogeneous setting, Fed-ALoRA updates all $A$ matrices locally, and transmits and aggregates only $B$ matrices on server side. In the heterogeneous setting, direct aggregation of $B$ is infeasible due to their distinct sizes, so we decompose $B$ into $(B_1, B_2)$ with appropriate sizes and introduce an auxiliary matrix for further dimension adjustment. Compared to full LoRA aggregation, Fed-ALoRA reduces communicated parameters by up to 50% and 75% in the homogeneous and heterogeneous settings, respectively, while maintaining performance.

- We conduct extensive experiments on intra-domain multi-task benchmarks such as commonsense reasoning and math reasoning, cross-domain multi-task NLP dataset, and federated NLP dataset to evaluate the effectiveness of our approaches. Across all datasets, our methods consistently deliver more balanced performance with comparable or superior accuracy compared to existing methods. In particular, **ALoRA** surpasses the sharing-$A$ approach **HydraLoRA**, improving average ROUGE-1 by +0.68 with a $\Delta m\%$ (which quantifies performance balance via mean drop from single-objective baselines) gain of -1.94. Similarly, **Fed-ALoRA** outperforms the sharing-$A$ approach **FedSA-LoRA**, achieving gains of +1.26 (homogeneous) and +1.96 (heterogeneous) with $\Delta m\%$ gains of -2.08 and -2.65, respectively. Compared with approaches that aggregate full LoRA parameters, our method attains comparable performance, smaller $\Delta m\%$, and substantially reduced communication cost by transmitting much fewer parameters.

## 2 BACKGROUND

### 2.1 LOW-RANK ADAPTATION

Pre-trained language models exhibit low intrinsic dimensionality when adapt to downstream tasks (Aghajanyan et al., 2021). LoRA leverages this property by approximating weight updates through low-rank decomposition. Particularly, for a pre-trained weight matrix $W_0 \in \mathbb{R}^{d_{\text{out}} \times d_{\text{in}}}$, the weight updates is defined as $\Delta W = BA$, where $A \in \mathbb{R}^{r \times d_{\text{in}}}$, $B \in \mathbb{R}^{d_{\text{out}} \times r}$, and the rank $r \ll min(d_{\text{in}}, d_{\text{out}})$. During training, only $A$ and $B$ matrices are trainable. Hence, given the input $x \in \mathbb{R}^{d_{\text{in}}}$, the forward pass is expressed as: $y = y_0 + \Delta y = W_0 x + BAx$. In practice, $A$ is typically initialized using Kaiming Uniform (He et al., 2015), and $B$ is initialized as zero to ensure $\Delta W = \mathbf{0}$.

### 2.2 FINE-TUNING WITH MULTIPLE LORAS

Multiple LoRA-based methods extend vanilla LoRA with additional modules to improve adaptability across heterogeneous domains (Zi et al., 2023; Dettmers et al., 2023). In multi-task fine-tuning, they often use MoE designs where LoRAs act as dynamically routed experts (Luo et al., 2024; Huang et al., 2024), while in federated fine-tuning, they aim to balance personalization with shared knowledge aggregation (Raje et al., 2025; Zhang et al., 2025b). A common idea among these multi-LoRA approaches is to share the same matrix $A$, based on the observation that $A$ matrices from LoRAs trained on different tasks or clients are often highly similar. For example, HydraLoRA (Tian et al., 2024) employs a single $A$ matrix and multiple $B$ matrices to express the weight updates: $\Delta W = \sum_{i=1}^{n} w_i B_i A$, where $n$ is the number of $B$ matrices, $w_i$ is the gating score for each $B_i$. The federated multi-LoRA approach FedSA-LoRA (Guo et al., 2025) shares only the $A$ matrices for server aggregation, after which the server broadcasts the aggregated $A$ to all clients. The model update of client $i$ is given by:

$$\Delta W_i^t = B_i^t \bar{A}^t, \quad \bar{A}^t = \text{Agg}(A_1^{t-1}, \cdots, A_n^{t-1}),$$

where $n$ is the number of clients, and $t$ is the current communication round, and $\text{Agg}(\cdot)$ denotes an aggregation algorithm such as simple averaging.

## 3 REVISITING PARAMETER SHARING IN MULTI-LORA FINE-TUNING

As noted earlier, a common strategy for parameter sharing is to reuse the same matrix $A$ across multiple LoRA modules, with the goal of reducing the total number of parameters and enabling knowledge transfer. In this section, we systematically re-examine this approach through a series of controlled experiments. Full implementation details are provided in Appendix A.

### 3.1 SIMILARITY IN $A$ STEMS FROM SAME INITIALIZATION, NOT SHARED KNOWLEDGE

A primary motivation for sharing $A$ across LoRA modules is the observation that the matrices $A_i$ of different LoRAs often appear similar during training. However, upon closer examination, we find that this similarity largely arises from their common initialization rather than from the shared knowledge. We fine-tune the LLaMA2-7B model (Touvron et al., 2023)[1] separately on classification and summarization tasks from the Dolly-15K dataset (Conover et al., 2023), using either identical or different random seeds for $A$ initialization (leading to different initializations for the matrices $A_i$), and compare the resulting LoRA modules using principal angle-based similarity (Zhu & Knyazev, 2013), where a value of $1$ indicates complete similarity and $0$ indicates dissimilarity. The results are shown in Figure 1, and details of the similarity metric are discussed in Appendix A.1.

*Observation.* Figure 1(Left) shows that with the same initialization, $A_i$ matrices are highly similar. In contrast, Figure 1(Middle, Right) shows that with different initializations, $A_i$ matrices from either the same or different tasks exhibit little similarity, while $B_i$ matrices display relatively higher similarity. These results suggest that the $A$ is highly sensitive to random seeds rather than necessarily capturing shared knowledge, whereas $B$ is less affected.

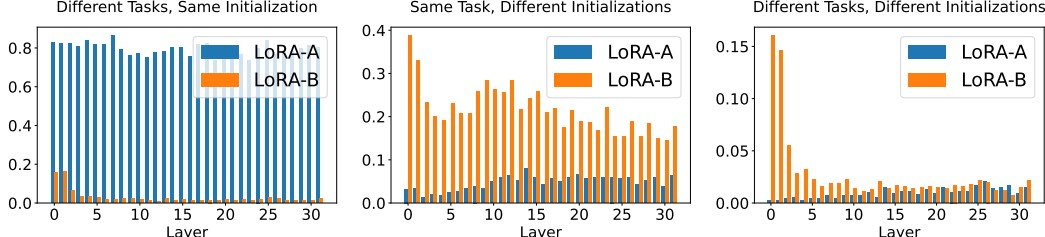

Figure 1: Layer-wise similarity analysis between different LoRA modules. Left: two different tasks with the same random seed. Middle: the same task with different random seeds. Right: two different tasks with different random seeds. $A_i$ matrices are similar only under the same initialization, whereas $B_i$ exhibits relatively stable similarity across different tasks and seeds.

### 3.2 DISSECTING DISTINCT DYNAMICS OF $A$ AND $B$ DURING TRAINING

The above analysis motivates us to further investigate the learning dynamics of $A$ and $B$ during training by comparing their states before and after fine-tuning on the summarization task[2]. Specifically, we examine the LoRA modules $\Delta W$ (see Section 2.2), $A$ and $B$. Our experiments evaluate (i) the similarity of modules $A$ and $B$ (using the similarity metric in Section 3.1) and (ii) the magnitude and directional variations of $\Delta W$, $A$ and $B$. To formalize this[3], any weight matrix $W \in \mathbb{R}^{d_{\text{out}} \times d_{\text{in}}}$ can be decomposed into a magnitude and a direction component: $W = \|W\|_c \frac{W}{\|W\|_c} = mV$, where $\|\cdot\|_c$ denotes the column-wise norm. Here, $m \in \mathbb{R}^{1 \times d_{\text{in}}}$ is the magnitude vector, with $m_j$ denoting the norm of the $j$-th column of $W$, and $V \in \mathbb{R}^{d_{\text{out}} \times d_{\text{in}}}$ is the direction matrix with unit-norm columns. Given two matrices $W_1$ and $W_2$, their magnitude and direction discrepancies are defined as

$$\Delta M = \frac{1}{d_{\text{in}}} \sum_{j=1}^{d_{\text{in}}} |m_{1,j} - m_{2,j}|, \qquad \Delta D = \frac{1}{d_{\text{in}}} \sum_{j=1}^{d_{\text{in}}} \big(1 - \cos(V_{1,j}, V_{2,j})\big).$$

---

[1]https://huggingface.co/meta-llama/Llama-2-7b

[2]We use the checkpoints from the second and final steps, since $B$ is initialized to zero at the beginning.

[3]We follow the same setup as in Liu et al. (2024).

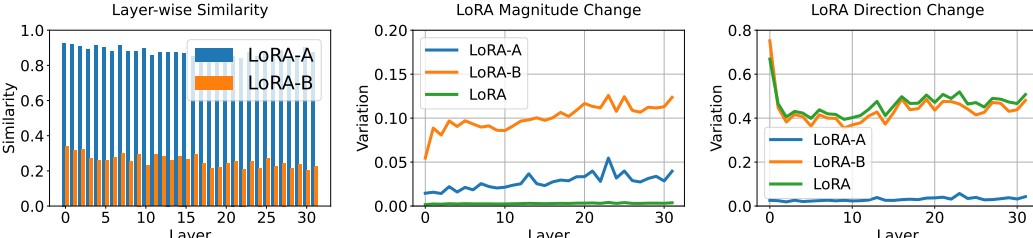

Figure 2: Comparison of LoRA modules before and after the fine-tuning. Left: similarity; Middle: magnitude change; Right: direction change. The module $A$ remains largely unchanged from initialization, whereas the module $B$ exhibits pronounced variation in both magnitude and direction. Overall, LoRA shows limited magnitude change, with nearly all directional change captured by $B$.

*Observation.* Figure 2(Left) shows that $A$ remains highly similar throughout training, undergoing only minimal changes, whereas $B$ exhibits much larger differences, indicating substantial adaptation after fine-tuning. Figures 2(Middle, Right) further reveal that the variations in $A$ are primarily in magnitude with little directional change, while $B$ accounts for most of the direction change. These results suggest that $A$ functions more as a fixed feature projector, whereas $B$ aggregates and adapts these features to encode domain knowledge. This highlights the more dominant role of $B$ over $A$ in knowledge learning, raising an intriguing question: might sharing the module $B$, rather than $A$, be more effective for parameter and knowledge sharing?

### 3.3 COMPARISON BETWEEN SHARING $A$ AND SHARING $B$

In this section, we address the question from Section 3.2 by comparing the performance of sharing modules $A$ and $B$ under both multi-task and federated fine-tuning.

**Gradient conflicts may lead to lazy learning for $A$ in multi-task fine-tuning**. Given an input $x \in \mathbb{R}^{d_{in}}$ and the gradient of the output $y$, $g \in \mathbb{R}^{d_{out}}$, the gradient of $A$ in the sharing-$A$ structure is $\nabla A = \sum_{i=1}^{n} w_i (B_i^\top g) x^\top$, where each term corresponds to a $B_i$ expert. We record the magnitudes of $\nabla A$ and compute the cosine similarity between gradient components, where negative similarity indicates a conflict that may hinder learning. The same procedure is applied to the sharing-$B$ structure. We then compare the two structures on the common-sense reasoning dataset (Hu et al., 2023), track-

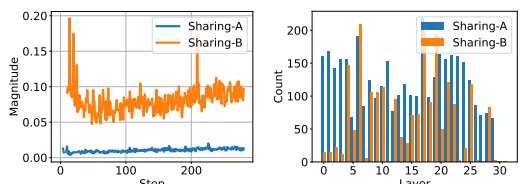

Figure 3: Comparing sharing $A$ versus $B$ in multi-task fine-tuning. Left: gradient magnitudes of $A$ and $B$. Right: number of gradient conflicts per layer. Sharing $A$ causes smaller gradient magnitudes and more frequent conflicts than sharing $B$.

ing both the gradient magnitudes of shared parameters and the number of conflicts throughout training. The results are shown in Figure 3.

*Observation.* Figure 3(Left) shows that the gradient magnitude of $A$ in the sharing-$A$ structure is near zero, while the gradient of $B$ in the sharing-$B$ structure is much larger. Figure 3(Right) shows that sharing $A$ also produces more gradient conflicts. Thus, in the sharing-$A$ structure, $A$ learns very slowly possibly due to the more frequent conflicting updates. We refer to this phenomenon as "lazy learning". Previous analysis in Section 3.2 indicates that $A$ functions as a feature projector. Hence, "lazy learning" may restrict the ability to explore diverse feature subspaces.

**Knowledge transfer in federated fine-tuning**. Each client fine-tunes its own LoRA and transmits the shared parameters to the server, which aggregates and returns them (full details can be found in Section 4.2). This setup allows us to assess whether the shared parameters improve knowledge transfer by evaluating each client's performance across all tasks. We compare the two structures across 8 clients, each assigned an NLP task from the FLAN dataset (Wei et al., 2022), and use ROUGE-1 score (Lin, 2004) to measure performance, where a value of 0 means no overlap between model prediction and ground truth, and 100 indicates perfect word-level overlap.

Table 1: Comparing sharing $A$ versus $B$ in federated fine-tuning. Sharing $B$ consistently outperforms sharing $A$ in both the homogeneous and heterogeneous settings.

| Setting | Method | Client 1 | Client 2 | Client 3 | Client 4 | Client 5 | Client 6 | Client 7 | Client 8 | Avg. |
|---|---|---|---|---|---|---|---|---|---|---|
| Homogeneous | Sharing $A$ | 50.02 | 49.78 | 59.92 | 42.66 | 54.05 | 25.21 | 23.32 | 49.43 | 44.30 |
| | Sharing $B$ | 67.43 | 67.83 | 69.85 | 69.26 | 69.50 | 60.94 | 54.86 | 70.90 | 66.32 |
| Heterogeneous | Sharing $A$ | 43.01 | 41.58 | 50.09 | 38.34 | 53.53 | 24.82 | 24.28 | 50.41 | 40.76 |
| | Sharing $B$ | 48.38 | 54.80 | 58.30 | 46.98 | 62.15 | 35.09 | 31.80 | 64.89 | 50.30 |

*Observation.* As shown in Table 1, in the homogeneous setting, sharing $B$ outperforms sharing $A$ by an average of 49.71%, with improvements ranging from 16.57% to 141.73%. In the heterogeneous setting, sharing $B$ achieves an average improvement of 23.41%, with gains ranging from 12.49% to 41.38%. These results clearly indicate that sharing $B$ better facilitates cross-client knowledge transfer than sharing $A$.

## 4 PROPOSED METHODS

Motivated by the findings in Section 3, we replace $A$ with $B$ as the shared parameter and propose two new multi-LoRA fine-tuning methods: **ALoRA** (Asymmetric LoRA) for multi-task training, and **Fed-ALoRA** for both homogeneous and heterogeneous federated settings.

### 4.1 ALoRA FOR MULTI-TASK FINE-TUNING

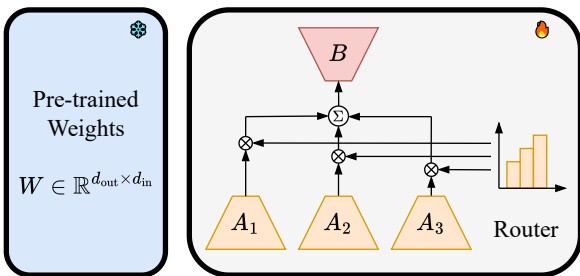

Figure 4: ALoRA adopts multiple $A$ and a single $B$ to explore diverse feature subspaces.

Multi-task fine-tuning typically adapts a pretrained LLM using data from multiple tasks. The goal is to improve generalization by learning from diverse inputs. The proposed ALoRA is illustrated in Figure 4. Given an input $x \in \mathbb{R}^{d_{\text{in}}}$, the forward pass is given by

$$y = y_0 + \Delta y = W_0 x + B \sum_{i=1}^{n} w_i A_i x,$$

where $W_0 \in \mathbb{R}^{d_{\text{out}} \times d_{\text{in}}}$ is the pre-trained weight matrix, $A_i \in \mathbb{R}^{r \times d_{\text{in}}}$ are the expert matrices, $B \in \mathbb{R}^{d_{\text{out}} \times r}$ is the shared aggregator, and the rank $r \ll min(d_{\text{in}}, d_{\text{out}})$. Each $A_i$ projects the input into a distinct feature subspace, and $B$ fuses the learned features to produce the output. The expert weights $w = (w_1, \ldots, w_n)$ are obtained from an input-aware router, implemented as a linear gating function with parameters $W_g \in \mathbb{R}^{n \times d_{\text{in}}}$: $w = \text{softmax}(W_g x)$.

During the inference, the router computes input-dependent weights, and the weighted average of the adapters is dynamically merged into the pre-trained weights.

### 4.2 FED-ALORA FOR FEDERATED FINE-TUNING

Federated fine-tuning can be divided into two settings: (i) *homogeneous*, where all clients adopt the same configuration, and (ii) *heterogeneous*, where clients have varying capacities, introducing both computational and communication heterogeneity.

**Homogeneous setting**. In this case, all $n$ clients fine-tune their LoRA modules with the same rank. Each update takes the form $\Delta W_i = B_i A_i$, where $A_i \in \mathbb{R}^{r \times d_{\text{in}}}$ and $B_i \in \mathbb{R}^{d_{\text{out}} \times r}$.

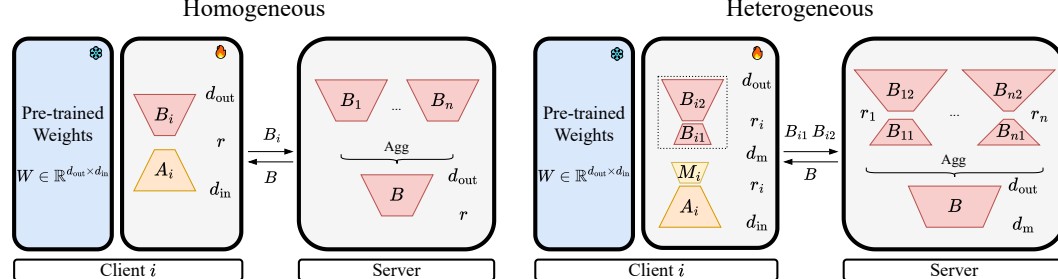

Figure 5: Fed-ALoRA shares only $B$ matrices for server aggregation. Left: Homogeneous setting (same rank), where the shared $B$ is directly transmitted. Right: Heterogeneous setting (different ranks), where the shared $B$ is decomposed into two matrices for heterogeneity. Compared to the standard full LoRA aggregation, the communication cost per client is reduced to $\mathcal{O}(d_{\text{out}}r)$ in the homogeneous setting and $\mathcal{O}(d_{\text{out}}r_i)$ in the heterogeneous setting if $d_m$ is chosen appropriately.

Figure 5(Left) illustrates the procedure of Fed-ALoRA for homogeneous setting, with the detailed steps for each communication round $t$ shown below ($t \geq 1$):

Step 1: **Initialization.** If $t = 1$, each client initializes $A_i$ randomly and sets $B_i$ to zero. For $t > 1$, $A_i$ and $B_i$ are initialized with $A_i^{t-1}$ and $B_0^{t-1}$, respectively.

Step 2: **Local training.** Each client performs LoRA fine-tuning on its local data, obtaining $(A_i^t, B_i^t)$ by optimizing $\mathcal{L}(W_0 + B_i A_i)$ with respect to $(A_i, B_i)$, where $\mathcal{L}(\cdot)$ is the loss function. The client then uploads only $B_i^t$ to the server for aggregation.

Step 3: **Aggregation.** The server aggregates the uploaded matrices using the operator $\text{Agg}(\cdot)$ from McMahan et al. (2017), and obtains $B_0^t \leftarrow \text{Agg}(B_1^t, \cdots, B_n^t)$.

Step 4: **Broadcast.** The server then sends the global matrix $B_0^t$ back to all clients.

*Remark.* In the full-LoRA aggregation, each client communicates $(d_{\text{in}} + d_{\text{out}})r$ parameters per round. In contrast, Fed-ALoRA requires transmitting only $B_i$, reducing the communication cost to $d_{\text{out}}r$.

**Heterogeneous setting**. In this case, clients may have different capacity constraints, resulting in parameterizations with diverse ranks $r_i$, given by $\Delta W_i = B_i A_i$, where $A_i \in \mathbb{R}^{r_i \times d_{\text{in}}}$ and $B_i \in \mathbb{R}^{d_{\text{out}} \times r_i}$ for $i = 1, \ldots, n$. Because the ranks differ across clients, direct averaging of the $B_i$ matrices is infeasible, and the aggregation strategy used in the homogeneous setting cannot be applied.

To address this issue, we propose a novel decomposition strategy of the form:

$$\Delta W_i = B_{i2} B_{i1} M_i A_i,$$

where $A_i \in \mathbb{R}^{r_i \times d_{\text{in}}}$, $M_i \in \mathbb{R}^{d_m \times r_i}$, $B_{i1} \in \mathbb{R}^{r_i \times d_m}$ and $B_{i2} \in \mathbb{R}^{d_{\text{out}} \times r_i}$. The high-level idea is to decompose the matrix $B_i$ of the same dimension into two components, $(B_{i1}, B_{i2})$, each of rank $r_i$. We further introduce $M_i$ as an intermediate matrix to control the dimension $d_m$. In addition, every client maintains an accumulator $B_{i0} \in \mathbb{R}^{d_{\text{out}} \times d_m}$ which stores the global updates it has received so far. The full procedure of round $t$ is illustrated in Figure 5(Right) and detailed below:

Step 1: **Initialization.** If $t = 1$, each client initializes $(A_i, M_i, B_{i1})$ randomly, and sets $(B_{i0}, B_{i2})$ to zero. For $t > 1$, $(A_i, M_i)$ is initialized with $(A_i^{t-1}, M_i^{t-1})$, $B_{i0}$ is initialzed with $B_0^{t-1}$, $B_{i1}$ is re-initialized randomly, and $B_{i2}$ is resets to zero.

Step 2: **Local training.** Each client performs LoRA fine-tuning on its local data, and obtains parameters $(A_i^t, M_i^t, B_{i1}^t, B_{i2}^t)$ by optimizing $\mathcal{L}(W_0 + (B_{i0} + B_{i2} B_{i1}) M_i A_i)$ with respect to $(A_i, M_i, B_{i1}, B_{i2})$. The client then uploads $(B_{i1}^t, B_{i2}^t)$ to the server.

Step 3: **Aggregation.** The server reconstructs $B_i^t = B_{i2}^t B_{i1}^t$ for each client and then performs the aggregation $B_0^t \leftarrow \text{Agg}(B_1^t, \cdots, B_n^t)$.

Step 4: **Broadcast.** The server then sends the global matrix $B_0^t$ back to all clients.

*Remark.* The previous parameter-sharing approach, FedSA-LoRA (Guo et al., 2025), does not support the heterogeneous setting. Fed-ALoRA addresses this limitation by introducing the decomposition $(B_{i1}, B_{i2})$, enabling efficient aggregation across clients with different capacities.

Table 2: Results on intra-domain multi-task commonsense reasoning benchmark. $\Delta m\%$ measures performance balance across tasks. $\downarrow$ denotes that lower values are better. All methods use the same number of adapter parameters. We independently run each experiment 3 times and report the mean and standard error.

| Method | ARC-C | ARC-E | BoolQ | HellaS. | OBQA | PIQA | SIQA | WinoG. | Avg. | $\Delta m\%(\downarrow)$ |
|---|---|---|---|---|---|---|---|---|---|---|
| Single | $77.76 \pm 0.97$ | $90.81 \pm 0.23$ | $73.39 \pm 1.05$ | $95.40 \pm 0.08$ | $86.60 \pm 0.72$ | $89.25 \pm 0.47$ | $80.69 \pm 0.48$ | $85.35 \pm 0.64$ | $84.90 \pm 0.21$ | |
| LoRA | $76.66 \pm 1.02$ | $88.72 \pm 0.83$ | $72.73 \pm 1.24$ | $94.50 \pm 0.41$ | $84.10 \pm 0.14$ | $87.59 \pm 0.15$ | $79.42 \pm 0.15$ | $85.40 \pm 1.00$ | $83.64 \pm 0.27$ | 1.48 |
| LoHa | $77.13 \pm 0.13$ | $88.91 \pm 0.09$ | $72.89 \pm 0.19$ | $94.05 \pm 0.23$ | $85.40 \pm 0.56$ | $87.84 \pm 0.34$ | $78.92 \pm 0.44$ | $84.73 \pm 0.17$ | $83.73 \pm 0.11$ | 1.36 |
| AdaLoRA | $77.72 \pm 0.72$ | $89.72 \pm 0.96$ | $73.32 \pm 0.19$ | $93.98 \pm 0.73$ | $85.20 \pm 1.13$ | $87.90 \pm 0.35$ | $79.45 \pm 0.18$ | $83.78 \pm 0.28$ | $83.88 \pm 0.32$ | 1.17 |
| MoSLoRA | $76.87 \pm 0.12$ | $89.22 \pm 0.15$ | $73.50 \pm 0.86$ | $95.02 \pm 0.13$ | $85.27 \pm 2.58$ | $87.99 \pm 0.59$ | $80.89 \pm 0.12$ | $85.13 \pm 0.55$ | $84.23 \pm 0.39$ | 0.76 |
| HydraLoRA | $78.58 \pm 0.12$ | $89.94 \pm 0.18$ | $75.02 \pm 0.18$ | $95.21 \pm 0.11$ | $84.90 \pm 1.55$ | $88.03 \pm 0.15$ | $79.99 \pm 0.72$ | $84.92 \pm 0.34$ | $84.57 \pm 0.17$ | 0.32 |
| ALoRA (ours) | $79.40 \pm 0.41$ | $89.69 \pm 0.71$ | $74.38 \pm 0.10$ | $94.85 \pm 0.23$ | $86.10 \pm 0.14$ | $88.24 \pm 0.49$ | $80.17 \pm 0.32$ | $85.68 \pm 0.27$ | $\mathbf{84.81} \pm 0.03$ | 0.04 |

*Remark.* In the vanilla full-LoRA aggregation, each clients uploads $(d_{\mathrm{in}} + d_{\mathrm{out}})r_i$ parameters to the server. The server then extends all heterogeneous updates to the maximum rank $r_{\mathrm{max}} = \max\{r_1, \cdots, r_n\}$ by padding with zeros, and broadcasts $(d_{\mathrm{in}} + d_{\mathrm{out}})r_{\mathrm{max}}$ parameters back to every client. In Fed-ALoRA, if $d_m$ is chosen comparable to $r_{\mathrm{max}}$ with $d_m \ll \min(d_{\mathrm{in}}, d_{\mathrm{out}})$, then client $i$ maintains $(d_{\mathrm{in}} + d_{\mathrm{out}} + 2d_m)r_i \approx (d_{\mathrm{in}} + d_{\mathrm{out}})r_i$ trainable parameters. The communication cost is reduced to $\mathcal{O}(d_{\mathrm{out}}r_i)$ from client to server, and $\mathcal{O}(d_{\mathrm{out}}r_{\mathrm{max}})$ from server to clients.

## 5 EXPERIMENTS

### 5.1 MULTI-TASK FINE-TUNING

We fine-tune the LLaMA3-8B model (Dubey et al., 2024)[4] on the *intra-domain* multi-task benchmark commonsense reasoning (Hu et al., 2023), which contains 8 question answering (QA) datasets, each focusing on a different aspect of commonsense. We also fine-tune the LLaMA2-7B model (Touvron et al., 2023)[5] on the *cross-domain* multi-task NLP dataset (Long et al., 2024), which mixes 8 different tasks such as QA, classification, and text generalization. These tasks are sampled from the FLAN dataset (Wei et al., 2022). For each task, we first fine-tune LoRA on its own dataset and use the performance as the single-task baseline. We then compare the proposed ALoRA with several representative methods: the vanilla LoRA (Hu et al., 2022), LoHa (Yeh et al., 2023), AdaLoRA (Zhang et al., 2023), MoSLoRA (Wu et al., 2024a), and HydraLoRA (Tian et al., 2024). We also examine the math reasoning benchmark (Hu et al., 2023). Full details and further discussion are provided in Appendix B. We also provide additional comparisons between HydraLoRA and ALoRA on more models in Appendix B.

To evaluate performance, we use the following metrics: (1) average accuracy for commonsense reasoning and average ROUGE-1 score for multi-task NLP dataset; and (2) $\Delta m\%$ (Maninis et al., 2019), the average per-task performance change against the single-task baseline. $\Delta m\% = \frac{1}{K}\sum_{k=1}^{K}(-1)^{\delta_k}(M_k - M_0)/M_0 \times 100$, where $M_k$ is the performance of $k$-th task under the compared method, $M_0$ is the baseline performance. $\delta_k = 1$ if higher values indicate better performance, otherwise $\delta_k = 0$. This metric evaluates how well performance is balanced across multiple tasks.

The results are presented in Tables 2 and 3. ALoRA achieves slightly better average accuracy than existing LoRA variants with the most balanced results in both benchmarks. In commonsense reasoning, the hardest task is ARC-C, and ALoRA is the only method that exceeds the single-task baseline. In multi-task NLP dataset, the most challenging task is summarization (Sum), where ALoRA effectively mitigates negative influence from other tasks and achieves the best performance on this task. These results suggest that ALoRA encourages knowledge transfer.

### 5.2 FEDERATED FINE-TUNING

We fine-tune the LLaMA2-7B model and evaluate on the federated dataset constructed by Long et al. (2024), which includes 8 NLP tasks sampled from FLAN dataset (Wei et al., 2022), with each client assigned to one task. For each client, we first fine-tune LoRA on its own training dataset, and use the performance as the single-client baseline. We use ROUGE-1 as the evaluation metric.

In the homogeneous setting, we compare our Fed-ALoRA with: FedIT (Zhang et al., 2024), FedDPA (Long et al., 2024), and FedSA-LoRA (Guo et al., 2025). In the heterogeneous setting, we compare

---

[4] https://huggingface.co/meta-llama/Meta-Llama-3-8B
[5] https://huggingface.co/meta-llama/Llama-2-7b

Table 3: Results on cross-domain multi-task NLP datasets. $\Delta m\%$ measures performance balance across tasks. All methods use the same number of adapter parameters.

| Method | CSR | Ent | ODQA | Para | RC | Sent | Sum | TFmt | Avg. | $\Delta m\%(\downarrow)$ |
|---|---|---|---|---|---|---|---|---|---|---|
| Single | 45.15 | 65.00 | 75.19 | 55.00 | 78.00 | 71.75 | 28.17 | 88.6 | 63.36 | |
| LoRA | 53.19 | 63.00 | 84.31 | 51.00 | 50.50 | 69.50 | 32.53 | 89.36 | 61.67 | 0.31 |
| LoHa | 49.94 | 60.94 | 79.78 | 67.70 | 69.57 | 73.50 | 33.33 | 90.85 | 65.70 | -5.76 |
| AdaLoRA | 51.94 | 56.94 | 78.57 | 65.22 | 66.61 | 59.50 | 31.95 | 84.93 | 61.96 | -0.41 |
| MoSLoRA | 50.70 | 60.50 | 81.11 | 71.50 | 70.00 | 75.00 | 32.64 | 87.95 | 66.18 | -6.58 |
| HydraLoRA | 44.51 | 67.50 | 75.83 | 74.50 | 76.50 | 71.50 | 32.14 | 89.10 | 66.45 | -6.39 |
| ALoRA (ours) | 48.21 | 62.50 | 80.35 | 78.50 | 68.50 | 75.00 | 33.79 | 90.20 | **67.13** | **-8.33** |

Table 4: Results for the **homogeneou**s federated setting. Params.(M) denotes the average number of parameters (in millions) transmitted per client in each round. ALoRA achieves the most balanced performance while reducing communication cost by 50% compared to full LoRA aggregation FedIT.

| Method | Coref | Ent | LAcc | Para | QCls | S2T | TFmt | WSD | Avg. | $\Delta m\%(\downarrow)$ | Params.(M) |
|---|---|---|---|---|---|---|---|---|---|---|---|
| Single | 73.00 | 84.00 | 79.00 | 78.00 | 94.00 | 72.21 | 96.64 | 60.50 | 79.67 | | |
| FedIT | 86.24 | 86.50 | 78.00 | 81.00 | 94.50 | 72.06 | 96.51 | 65.00 | 82.47 | -3.92 | 8.39 |
| FedDPA | 88.51 | 85.50 | 73.50 | 77.50 | 95.50 | 73.76 | 96.40 | 65.00 | 81.96 | -3.30 | 16.78 |
| FedSA-LoRA | 81.77 | 86.00 | 78.00 | 75.00 | 93.50 | 73.34 | 96.55 | 65.00 | 81.15 | -2.21 | **4.19** |
| Fed-ALoRA (ours) | 85.74 | 87.00 | 73.50 | 79.00 | 94.00 | 73.10 | 96.24 | 71.50 | **82.51** | **-4.29** | **4.19** |

Fed-ALoRA with: ZeroPadding, FLoRA (Wang et al., 2024), and FedSA-LoRA (Guo et al., 2025), which does not natively support heterogeneity but is adapted here using the decomposition proposed in our method. We also report the average number of parameters communicated per client in each round, including both uploads to the server and downloads from the server. In the homogeneous setting, all clients use rank 8. In the heterogeneous setting, the ranks are $\{64, 64, 32, 32, 16, 16, 8, 8\}$, with $d_m$ set to 16. Full details are provided in the Appendix B.

The results are presented in Table 4 and Table 5. Fed-ALoRA achieves the most balanced performance while reducing communication cost by 50% compared to full-LoRA aggregation in homogeneous setting, and reducing by 75% in heterogeneous setting. Notably, the client with the word sense disambiguation (WSD) task performs poorly. However, Fed-ALoRA outperforms both the single-client baseline and full-LoRA aggregation in homogeneous setting and ranks second in heterogeneous setting. These results show that Fed-ALoRA effectively promotes knowledge sharing across clients.

Table 5: Results for the **heterogeneous** setting. ALoRA achieves the most balanced performance while reducing communication cost by 75% compared to full LoRA aggregation ZeroPadding. The original FedSA-LoRA does not support heterogeneity; $*$ denotes implementation with our decomposition strategy.

| Method | Coref | Ent | LAcc | Para | QCls | S2T | TFmt | WSD | Avg. | $\Delta m\%(\downarrow)$ | Params.(M) |
|---|---|---|---|---|---|---|---|---|---|---|---|
| Single | 81.62 | 88.00 | 81.00 | 79.50 | 94.50 | 72.07 | 96.64 | 60.50 | 81.73 | | |
| ZeroPadding | 86.95 | 87.00 | 77.50 | 79.50 | 94.00 | 72.87 | 96.46 | 64.00 | 82.29 | -0.91 | 49.28 |
| FLoRA | 82.03 | 87.50 | 75.50 | 74.00 | 95.50 | 70.99 | 96.07 | 62.00 | 80.45 | 1.54 | 141.56 |
| FedSA-LoRA$^*$ | 80.26 | 83.50 | 76.50 | 76.00 | 93.00 | 73.49 | 96.61 | 54.00 | 79.17 | 3.39 | **12.12** |
| Fed-ALoRA (ours) | 88.27 | 89.50 | 79.50 | 77.00 | 94.00 | 72.35 | 96.37 | 63.00 | **82.50** | **-1.07** | **12.12** |

## 5.3 IN-DEPTH ANALYSIS

**Multi-task fine-tuning**. We analyze the gate activations of HydraLoRA and ALoRA during inference on the commonsense reasoning benchmark. Figure 6 presents the t-SNE visualization of the gate activations in the last layer. The results show that ALoRA activations form clearer clusters than HydraLoRA. Some tasks share similar gate activations, suggesting that they prefer the same $A$ experts. This indicates that ALoRA is more effective at capturing diverse feature subspaces across tasks. In contrast, HydraLoRA relies on a single $A$ matrix, which limits its ability to explore diverse feature subspaces and leads to more scattered activations.

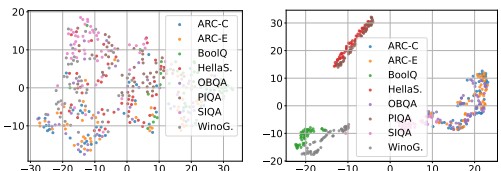

Figure 6: Gate activations of HydraLoRA and ALoRA. ALoRA yields more distinct clusters.

Table 6: Results of different intermediate ranks in the heterogeneous setting.

| Fed-ALoRA | Avg. | $\Delta m\%(\downarrow)$ |
|-----------|------|--------------------------|
| $d_m = 8$ | 81.92 | -0.41 |
| $d_m = 16$ | **82.50** | **-1.07** |
| $d_m = 32$ | 82.25 | -0.80 |
| $d_m = 64$ | 82.37 | -1.01 |

**Federated fine-tuning**. To further evaluate knowledge sharing, we compare the performance of each client on all tasks between Fed-ALoRA and FedSA-LoRA. This allows us to assess which parameters should be shared to improve cross-client transfer. As shown in Table 1, Fed-ALoRA achieves better results. In addition, we study the effect of different choices of the intermediate rank $d_m$ in the heterogeneous setting. The results in Table 6 show that with a proper choice of $d_m$, we can reduce the communication cost while maintaining the performance balance.

## 6 RELATED WORK

**Low-rank adaptation**. Vanilla LoRA (Hu et al., 2022) reparameterizes weight updates using low-rank matrices, enabling efficient fine-tuning without extra inference latency. Extensions develop along three directions. For rank allocation, AdaLoRA (Zhang et al., 2023) prunes less important singular values, and DyLoRA (Valipour et al., 2023) trains LoRA blocks with different ranks for flexible inference. For memory efficiency, QLoRA (Dettmers et al., 2023) applies 4-bit quantization, and SparseLoRA (Khaki et al., 2025) updates only a sparse subset of parameters using SVD. For structural variation, LoHa and LoKr (Yeh et al., 2023) adopt Hadamard and Kronecker decompositions, and DoRA (Liu et al., 2024) separates magnitude and direction. This paper provides a deep investigation into the training dynamics of modules $A$ and $B$, demonstrating the more dominant role of $B$ in knowledge learning and transfer.

**Multi-task fine-tuning**. Fine-tuning on multiple tasks improves generalization and transfer. A popular idea is to integrate LoRA with MoE, where experts specialize in different tasks. Among them, LoRAMoE (Dou et al., 2024), MoELoRA (Luo et al., 2024) and MoRE (Zhang et al., 2025a) align experts with task information to balance performance. SMoRA (Zhao et al., 2025) treats each rank as an expert, and ThanoRA (Liang et al., 2025) builds task-aware LoRA modules. DynMoLE (Li et al., 2025) uses entropy-based routing, and HydraLoRA (Tian et al., 2024) improves parameter efficiency by sharing $A$ matrices. In contrast to HydraLoRA, our proposed **ALoRA** shares matrices $B$, which promotes diverse feature projections and facilitates more effective knowledge transfer.

**Federated fine-tuning**. The models are adapted across clients while preserving data privacy. Existing methods fall into homogeneous and heterogeneous settings. In the homogeneous case, FedIT (Zhang et al., 2024) aggregates full LoRA parameters, while FedSA-LoRA (Guo et al., 2025) reduces communication by sharing only $A$. In the heterogeneous case, HetLoRA (Cho et al., 2024) supports varying ranks via self-pruning with sparse aggregation, Ravan (Raje et al., 2025) introduces multi-head LoRA updates, and FedALT (Bian et al., 2025) employs MoE-based adapters; FLoRA (Wang et al., 2024) provides a unified stacking framework. Unlike FedSA-LoRA, which is restricted to homogeneous ranks, our **Fed-ALoRA** shares $B$, reducing communication costs while supporting heterogeneous ranks through a decomposition strategy.

## 7 CONCLUSION

Our study shows that the similarity of LoRAs' $A$ matrices arises mainly from initialization rather than shared knowledge, with $B$ serving as the key component for knowledge transfer. Building on this insight, we propose ALoRA and Fed-ALoRA, which share $B$ for multi-task and federated fine-tuning. Experiments across diverse benchmarks demonstrate that these methods achieve more balanced performance while maintaining or improving accuracy over existing multi-LoRA approaches. Future work will further examine the distinct learning dynamics of $A$ and $B$ and develop new fine-tuning strategies inspired by these insights.

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

# SUPPLEMENTARY MATERIALS

## A  ADDITIONAL DETAILS FOR SECTION 3

### A.1  SIMILARITY METRIC IN LoRA MODULES

LoRA represents the weight updates as $\Delta W = BA$ with $A \in \mathbb{R}^{r \times d_{\text{in}}}$, $B \in \mathbb{R}^{d_{\text{out}} \times r}$. For any invertible $R \in \mathbb{R}^{r \times r}$, we have

$$\Delta W = BA = (BR)(R^{-1}A).$$

This shows that $A$ and $B$ are not individually unique. They can be arbitrarily rotated within the rank-$r$ subspace without changing $\Delta W$. As a result, directly computing the cosine similarity between $A$ or $B$ matrices can give misleading results.

Different seeds or initializations may lead to very different $A$ and $B$, but the subspaces they span are rotation-invariant. If the subspaces align, the modules are functionally aligned. Therefore, we use the subspace similarity proposed by Zhu & Knyazev (2013). Specifically, given two matrices $M_1, M_2 \in \mathbb{R}^{d \times r}$, we compute SVD of each and obtain the orthonormal bases of their column spaces, $U_1, U_2 \in \mathbb{R}^{r \times r}$. The similarity is then defined as

$$\text{Sim}(M_1, M_2) = \frac{1}{r}\|U_1^\top U_2\|_F^2 \in [0, 1],$$

where a higher value indicates a stronger alignment.

### A.2  ANALYSIS IN FEDERATED FINE-TUNING

Section 3.1-3.2 analyze the learning behavior of $A$ and $B$ matrices during fine-tuning on different tasks. We also perform the same analysis in the federated fine-tuning with two clients. Following Zhang et al. (2024), we randomly sample data from the Dolly-15K dataset (Conover et al., 2023) and split them into two clients, each containing 1493 instruction data samples from different tasks. The data distribution is shown in Figure 7.

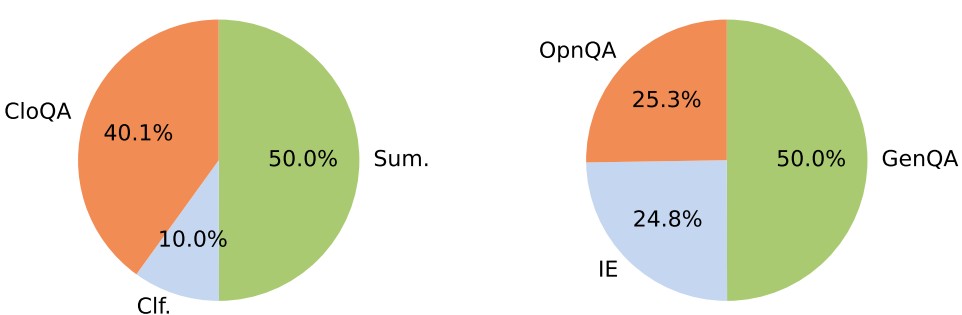

Figure 7: Data distribution of clients. Left: Client 1 contains Closed QA, Summarization, and Classification tasks. Right: Client 2 contains Open QA, General QA, and Information Extraction tasks.

We fine-tune the LLaMA2-7B model on the two clients and analyze how the similarity of LoRA modules is affected by random seeds for initialization. Using the similarity metric described in Appendix A.1, we compute the layer-wise similarity of LoRA modules across the two clients. The results, shown in Figure 8, indicate that, contrary to the assumption in FedSA-LoRA (Guo et al., 2025), the similarity of $A$ matrices across clients comes mainly from identical initialization rather than shared knowledge.

Furthermore, we analyze the learning dynamics of $A$ and $B$ in the federated fine-tuning setting. On client 2, we compute the similarity of $A$ before and after fine-tuning, and do the same for $B$. We also calculate the magnitude and direction variation for this client. The results, shown in Figure 9, are consistent with our earlier fine-tuning experiments on different tasks. They confirm that $B$ plays a more critical role than $A$ in encoding knowledge across clients.

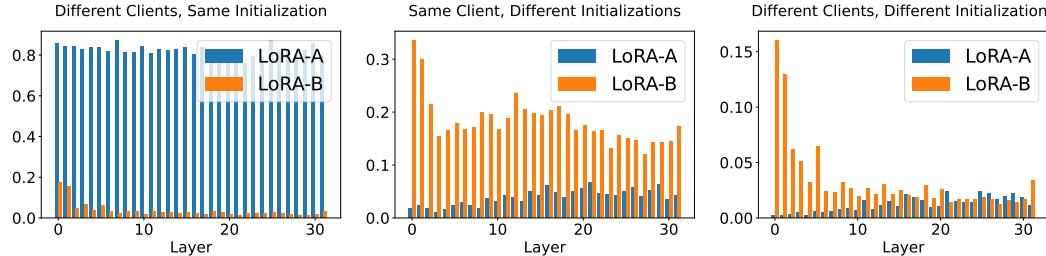

Figure 8: Layer-wise similarity analysis of LoRA modules across clients in federated fine-tuning. Left: two different clients with the same random seed. Middle: the same client with different random seeds. Right: two different clients with different random seeds. $A_i$ matrices are similar only under the same initialization, whereas $B_i$ exhibits relatively stable similarity across different clients and seeds.

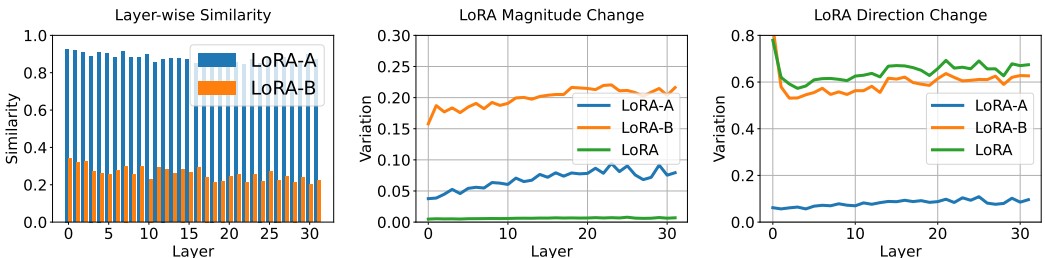

Figure 9: Comparison of LoRA modules on each client before and after federated fine-tuning. Left: similarity; Middle: magnitude change; Right: direction change. LoRA shows limited magnitude change, with nearly all directional change captured by $B$.

To further explore whether the above observation depends on the model or dataset, we analyze the learning dynamics of LoRA the Qwen2-7B (Yang et al., 2025)[6] model using the bigscience/xP3 dataset (Muennighoff et al., 2023), which contains data from 46 languages and 16 NLP tasks. We sample 3,000 English examples and fine-tune the model using the same configuration. The results are shown in Figure 10. We observe the same pattern: the $B$ matrix plays a more dominant role than $A$ matrix during training.

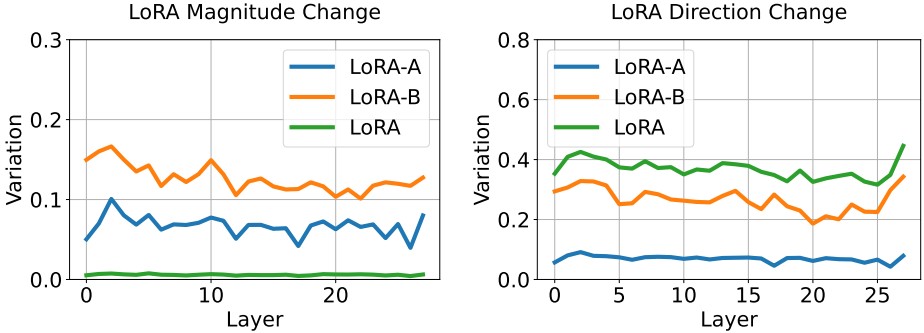

Figure 10: Comparison of LoRA modules using Qwen2-7B before and after federated fine-tuning. Left: magnitude change; Right: direction change.

## A.3 PRACTICAL IMPLEMENTATION

For the results in Section 3.1–3.2, we fine-tune the LLaMA2-7B model on data sampled from the Dolly-15K dataset for 3 epochs. The training uses a learning rate of 3e-4, batch size 32, and gradient accumulation step 2. We follow the alpaca_short template (Taori et al., 2023) to construct the instruction data. LoRA is applied to the $q_{\text{proj}}$ modules with rank $r = 8$.

---

[6] https://huggingface.co/Qwen/Qwen2-7B

For the analysis of lazy learning in multi-task fine-tuning (Section 3.3), we fine-tune the LLaMA3-8B model on the commonsense reasoning 15K dataset (Hu et al., 2023) for 3 epochs, using a learning rate of 3e-4, batch size 4, and gradient accumulation step 4. LoRA is applied to the $q_{\text{proj}}$ modules with rank $r = 8$. The sharing-$A$ structure uses 3 $A$ matrices, and the sharing-$B$ structure uses 3 $B$ matrices.

For the analysis of knowledge transfer in federated fine-tuning (Section 3.3), we fine-tune the LLaMA2-7B model on the federated NLP dataset (Long et al., 2024) for 10 communication rounds, with 10 local epochs per client. The learning rate is 5e-4, the batch size is 32, and the gradient accumulation step is 2. In the homogeneous setting, all clients use rank 8. In the heterogeneous setting, the ranks are set to $\{64, 64, 32, 32, 16, 16, 8, 8\}$. The methods are applied to $q_{\text{proj}}$ and $v_{\text{proj}}$ modules. All experiments are conducted on RTX A6000 GPU.

## B  ADDITIONAL EXPERIMENTAL DETAILS AND RESULTS

### B.1  BENCHMARKS

The intra-domain multi-task commonsense reasoning 170K benchmark contains questions from the following datasets: (1) ARC-Challenge and ARC-Easy (Clark et al., 2018), which consist of grade-school–level multiple-choice science questions; (2) BoolQ (Clark et al., 2019), a yes/no question-answering dataset requiring non-factoid reasoning and entailment; (3) HellaSwag (Zellers et al., 2019), a dataset of commonsense natural language inference (NLI) questions that require identifying the most appropriate continuation of a narrative input; (4) OpenBookQA (Mihaylov et al., 2018), which contains questions requiring multi-step reasoning by combining provided scientific facts with external background knowledge; (5) PIQA (Bisk et al., 2020), a dataset of everyday commonsense reasoning questions about the physical world; (6) SIQA (Sap et al., 2019), which focuses on social and emotional commonsense reasoning in everyday human interactions; (7) WinoGrande (Sakaguchi et al., 2021), a collection of fill-in-the-blank sentences designed to test pronoun resolution using commonsense.

The cross-domain multi-task NLP dataset contains 8 NLP tasks sampled from the FLAN dataset (Wei et al., 2022). The tasks are: (1) Commonsense, a reasoning task that requires everyday knowledge to make judgments; (2) Entailment, an NLI task that determines the relationship between a premise and a hypothesis; (3) Open-domain QA, a question answering task that retrieves or generates answers from open sources; (4) Paraphrase, a classification task that recognizes whether a sentence pair is semantically equivalent; (5) Reading comprehension, a question answering task requires understanding the text content and answering the related questions; (6) Sentiment classification, a classification task that determines the whether the sentiment polarity is neutral, positive, or negative; (7) Summarization, an NLG task that produces a compact digest of a long passage while keeping the critical information; (8) Text formatting, an NLG task that corrects the punctuation in unformatted text. Each task has 300 examples for training and 200 examples for testing.

The federated NLP dataset also contains 8 NLP tasks sampled from the FLAN dataset (Wei et al., 2022). The tasks are: (1) Coreference, a discourse understanding task that requires determining which entity a pronoun refers to; (2) Entailment, an NLI task that determines the relationship between a premise and a hypothesis; (3) Linguistic Acceptability, a classification task that detects whether a sentence is grammatical; (4) Paraphrase, a classification task that recognizes whether a sentence pair is semantically equivalent; (5) Question classification, a task for question understanding in question answering systems; (6) Structure-to-Text, a natural language generation (NLG) task that converts structured triples into natural language; (7) Text formatting, an NLG task that corrects the punctuation in unformatted text; (8) Word sense disambiguation, a classification task that determines whether the same word has the same meaning in two different sentences. Each task has 300 examples for training and 200 examples for testing, and we assign one task to each client.

For the intra-domain multi-task fine-tuning, we also consider the math reasoning 10K benchmark (Hu et al., 2023), which includes 4 datasets: (1) AQuA (Ling et al., 2017), which contains multiple-choice algebra word problems, each accompanied by a natural language rationale explaining the step-by-step reasoning; (2) GSM8K (Cobbe et al., 2021), a high-quality collection of linguistically diverse grade-school–level math word problems designed to evaluate multi-step reasoning; (3) MAWPS (Koncel-Kedziorski et al., 2016), a compilation of math word problems intended to support

robust and scalable research on arithmetic reasoning, including AddSub (basic addition/subtraction), SingleOp (single-operator arithmetic), MultiArith (multi-step arithmetic), SingleEq (single-equation algebra); (4) SVAMP (Patel et al., 2021), which consists of simple one-unknown grade-school–level arithmetic word problems, designed to test robustness in arithmetic reasoning.

The original split of the math reasoning 10K benchmark is not suitable for multi-task fine-tuning, since it does not include the full training data of the subsidiary tasks. In addition, Hu et al. (2023) report data leakage issues in this benchmark. To address these concerns, we downloaded the original data of each single task, and checked every training example in the benchmark to determine whether it belongs to the training set of any individual task. This process allowed us to construct a training dataset for each task, making it possible to fine-tune on single tasks and obtain single-task baselines. For multi-task fine-tuning, we fine-tune directly on the benchmark and evaluate on each task. Since the single-task training splits are created by us, we report the corresponding results in the Appendix.

## B.2 BASELINES

For multi-task fine-tuning, we compare our ALoRA with the following methods: (1) the vanilla LoRA (Hu et al., 2022); (2) LoHa (Yeh et al., 2023), which employs Hadamard decompositions to the updates; (3) AdaLoRA (Zhang et al., 2023), which adaptively prunes rank using SVD; (4) MoSLoRA (Wu et al., 2024a), which introduces an additional matrix to fuse update subspaces; (5) HydraLoRA (Tian et al., 2024), which adopts a sharing-$A$ multi-LoRA framework.

For homogeneous federated fine-tuning, we compare our Fed-ALoRA with the following methods: (1) FedIT (Zhang et al., 2024), where each client transmits the full LoRA parameters; (2) FedDPA (Long et al., 2024), which employs both a global adapter and a local adapter for each client; (3) FedSA-LoRA (Guo et al., 2025), which shares only the $A$ matrices. For the heterogeneous federated fine-tuning, we compare Fed-ALoRA with: (1) ZeroPadding, which pads all heterogeneous ranks to the maximum rank across clients, enabling FedIT to support heterogeneity; (2) FLoRA (Wang et al., 2024), a stacking-based noise-free aggregation method; (3) FedSA-LoRA (Guo et al., 2025), which does not natively support heterogeneity but is adapted here using the decomposition proposed in our method.

## B.3 PRACTICAL IMPLEMENTATION

For intra-domain multi-task fine-tuning on commonsense reasoning and math reasoning, we fine-tune the LLaMA3-8B model for 3 epochs on the training data with a learning rate of 3e-4. The AdamW optimizer is used with $\beta_1 = 0.9$ and $\beta_2 = 0.999$. The batch size is 4, and the gradient accumulation step is 4. For HydraLoRA, we follow the original setup with one single $A$ matrix and 3 $B$ matrices with rank 8. For our ALoRA, we use 3 $A$ matrices and a single $B$ matrix with rank 8. To ensure a fair comparison, the other baselines are configured with a comparable parameter size: LoRA, LoHa, and MoSLoRA use rank 16, while others use rank 8. All methods are applied to the $q_{\text{proj}}$ and $o_{\text{proj}}$ modules.

For cross-domain multi-task fine-tuning, we fine-tune the LLaMA2-7B model for 50 epochs on the training data with a learning rate of 3e-4. The AdamW optimizer is used with $\beta_1 = 0.9$ and $\beta_2 = 0.999$. The batch size is 4, and the gradient accumulation step is 4. For HydraLoRA, we follow the original setup with one single $A$ matrix and 3 $B$ matrices with rank 8. For our ALoRA, we use 3 $A$ matrices and a single $B$ matrix with rank 8. To ensure a fair comparison, the other baselines are configured with a comparable parameter size: LoRA, LoHa, and MoSLoRA use rank 16, while others use rank 8. All methods are applied to the $q_{\text{proj}}$ and $v_{\text{proj}}$ modules.

For federated fine-tuning, we fine-tune the LLaMA2-7B model for 10 communication rounds with 10 local epochs per client, using a learning rate of 5e-4. The AdamW optimizer is used with $\beta_1 = 0.9$ and $\beta_2 = 0.999$. The batch size is 32, and the gradient accumulation step is 2. In the homogeneous setting, all clients use rank 8. In the heterogeneous setting, the ranks are set to $\{64, 64, 32, 32, 16, 16, 8, 8\}$, and $d_m$ is chosen to 16. All methods are applied to $q_{\text{proj}}$ and $v_{\text{proj}}$ modules. All experiments are conducted on RTX A6000 GPU.

Table 7: Results on math reasoning. $\Delta m\%$ measures performance balance across tasks. ALoRA achieves the most balanced results.

| Method | AQuA | GSM8K | SVAMP | MAWPS | SingleEq | Avg. | $\Delta m\%(\downarrow)$ |
|---|---|---|---|---|---|---|---|
| Single | 28.34 | 63.68 | 71.10 | 86.13 | 90.75 | 68.00 | |
| LoRA | 30.31 | 66.26 | 75.10 | 90.34 | 94.88 | 71.38 | -5.21 |
| LoHa | 27.56 | 63.84 | 76.70 | 90.76 | 94.88 | 70.75 | -3.06 |
| AdaLoRA | 25.20 | 59.44 | 72.60 | 86.55 | 91.93 | 67.14 | 2.77 |
| MoSLoRA | 28.74 | 67.40 | 77.10 | 89.08 | 94.06 | 71.28 | -4.54 |
| HydraLoRA | 27.17 | 68.76 | 75.60 | 90.34 | 93.31 | 71.04 | -3.58 |
| ALoRA | 29.53 | 67.17 | 77.40 | 89.50 | 94.49 | **71.62** | **-5.31** |

Table 8: Results of different intermediate ranks in the heterogeneous setting.

| Fed-ALoRA | Coref | Ent | LAcc | Para | QCls | S2T | TFmt | WSD | Avg. | $\Delta m\%(\downarrow)$ |
|---|---|---|---|---|---|---|---|---|---|---|
| $d_m = 8$ | 90.75 | 84.50 | 79.50 | 73.50 | 95.00 | 73.05 | 96.09 | 63.00 | 81.92 | -0.41 |
| $d_m = 16$ | 88.27 | 89.50 | 79.50 | 77.00 | 94.00 | 72.35 | 96.37 | 63.00 | **82.50** | **-1.07** |
| $d_m = 32$ | 88.63 | 86.50 | 78.00 | 79.00 | 94.00 | 72.27 | 96.60 | 63.00 | 82.25 | -0.80 |
| $d_m = 64$ | 85.70 | 89.00 | 81.00 | 75.50 | 94.00 | 72.31 | 96.40 | 65.00 | 82.37 | -1.01 |

## B.4 ADDITIONAL EXPERIMENT RESULTS

The results on the math reasoning benchmark are presented in Table 7. LoRA, MoSLoRA, and our ALoRA outperform the single-task baseline on all tasks, but ALoRA achieves the most balanced performance, showing that it enables more effective knowledge transfer than the baselines. We also provide the full results of the study of different intermediate ranks in Section 5.3, which are shown in Table 8.

To further compare the effectiveness of sharing $A$ and sharing $B$, we provide an additional study of HydraLoRA and our ALoRA using Qwen2-7B (Yang et al., 2025)[7] and LLaMA2-13B (Touvron et al., 2023)[8]. We fine-tune the models using the same configuration as before, and report the mean and standard error over 3 independent runs for Qwen-7B. The results on LLaMA2-13B are less stable, likely because the configuration is suboptimal for the larger model, so we report only the best result from 3 runs. The comparisons shown in Table 9 validate that ALoRA consistently outperforms HydraLoRA.

Table 9: Comparison between HydraLoRA and ALoRA on intra-domain multi-task commonsense reasoning benchmark.

| Method | ARC-C | ARC-E | BoolQ | HellaS. | OBQA | PIQA | SIQA | WinoG. | Avg. |
|---|---|---|---|---|---|---|---|---|---|
| HydraLoRA (Qwen-7B) | 84.60 ± 0.30 | 93.20 ± 0.03 | 73.37 ± 1.42 | 94.42 ± 0.36 | 88.30 ± 0.42 | 89.58 ± 0.72 | 80.97 ± 0.06 | 84.33 ± 0.95 | 86.09 ± 0.42 |
| ALoRA (Qwen-7B) | 85.28 ± 0.42 | 93.69 ± 0.30 | 73.41 ± 0.15 | 94.76 ± 0.36 | 90.10 ± 0.42 | 89.07 ± 0.07 | 80.94 ± 0.47 | 84.50 ± 0.50 | **86.47 ± 0.07** |
| HydraLoRA (LLaMA2-13B) | 59.90 | 66.46 | 72.42 | 56.46 | 66.80 | 83.95 | 74.26 | 83.50 | 70.47 |
| ALoRA (LLaMA2-13B) | 74.40 | 86.03 | 74.43 | 67.62 | 82.80 | 79.87 | 80.71 | 76.96 | **77.86** |

## C THE USE OF LARGE LANGUAGE MODELS (LLMS)

In preparing this manuscript, large language models (LLMs) were used only to assist with language polishing and stylistic refinement. All technical content, formulations, experimental designs, and conceptual contributions were developed by the authors. Importantly, LLMs were not used for ideation and methodology development.

---

[7]https://huggingface.co/Qwen/Qwen2-7B
[8]https://huggingface.co/meta-llama/Llama-2-13b-hf

