# OpenReview forum: "Rethinking Parameter Sharing for LLM Fine-Tuning with Multiple LoRAs"
_ICLR.cc/2026/Conference — ICLR 2026 Conference Withdrawn Submission_

### Official Review · Reviewer_FJ1x · 2025-10-28

**Soundness:** 3
**Presentation:** 2
**Contribution:** 2
**Rating:** 4
**Confidence:** 4

**Summary:**

This paper investigated the mixture of expert setup in low-rank adaptation (LoRA) of LLMs, which are particularly useful in multi-task and federated learning. It is observed that similar $A$ matrices are due to the shared initialization rather than knowledge, and multiple $A$ matrices are prone to gradient conflicts leading to lazy learning. Consequently, the authors advocated ALoRA that relies on multiple $A$ matrices yet a single shared $B$. For federated learning, the $B$ matrix is shared by decomposing into two components to account for heterogeneous tasks. Experiments are conducted on four datasets to demonstrate the effectiveness of the design.

**Strengths:**

1. The three observations in Section 3 are interesting, which well motivates the design of ALoRA and Fed-ALoRA.
2. The methodology and design choices are clearly presented.
3. The experimental results showcase a consistent improvement over HydraLoRA and FedSA-LoRA.

**Weaknesses:**

1. My major concern is the similarity between this paper and [1], which not only has a closely related title, but also explores the same idea of "multiple A and a single B". A more detailed comparison with [1] is needed to clarify the distinctions and contributions of this work.
2. Several closely related baselines that employ MoE for multi-task fine-tuning are missing in both the introduction and experiments, including LoRAMoE [2], MTL-LoRA [3], and CoLA [4].
3. In the experimental evaluation, the models are limited to Llama2-7B and Llama3-8B. It would strengthen the paper to include results from additional large language models.

[1] Q. Dong et al., "MASA: Rethinking the Representational Bottleneck in LoRA with Multi-A Shared Adaptation," *arXiv preprint arXiv:2510.06005*, 2025.
[2] S. Dou et al., "LoRAMoE: Alleviating world knowledge forgetting in large language models via MoE-style plugin," in *ACL*, 2024.
[3] Y. Yang et al,, "MTL-LoRA: Low-rank adaptation for multi-task learning," in *AAAI*, 2025.
[4] Y. Zhou et al., "Cola: Collaborative low-rank adaptation," *arXiv preprint arXiv:2505.15471*, 2025.

**Questions:**

See weaknesses above.

---

> ### Author Response · Authors · 2025-11-28
> **Response to Reviewer FJ1x**
>
> We sincerely appreciate your reply. We are committed to addressing your concerns.
>
> **W1. My major concern is the similarity between this paper and MASA [1], which not only has a closely related title, but also explores the same idea of "multiple A and a single B". A more detailed comparison with [1] is needed to clarify the distinctions and contributions of this work.**
>
> **A**. The paper was uploaded to arXiv recently, and we were not aware of it before. It compares sharing $A$ and $B$ from an information bottleneck perspective and concludes that sharing $B$ enables more information transfer. Their method uses a cross-layer parameter-sharing design, which is different from ours. In contrast, we study why $A$ matrices appear similar and analyze the training dynamics of $A$ and $B$. We conduct extensive empirical experiments showing that sharing $B$ is more effective for knowledge transfer. Our method focuses on multi-task and federated fine-tuning.
>
> **W2. Several closely related baselines that employ MoE for multi-task fine-tuning are missing in both the introduction and experiments, including LoRAMoE [2], MTL-LoRA [3], and CoLA [4].**
>
> **A**. We will include and discuss these methods in our later revision. We are also attempting to run some of them, such as MTL-LoRA [3], using the same configuration as our experiments. However, we have not yet been able to reproduce the results reported in the original paper. Once we obtain comparable results, we will include them in the revised version. The reported results in the original paper used different configurations, so we can't directly cite their results here.
>
>
> **W3. In the experimental evaluation, the models are limited to Llama2-7B and Llama3-8B. It would strengthen the paper to include results from additional large language models.**
>
> **A**. We conduct additional experiments using Qwen2-7B and LLaMA2-13B on commonsense reasoning. We fine-tune the models using the same configuration, and report the mean and standard error over 3 independent runs for Qwen-7B. The results on LLaMA2-13B are less stable, likely because the configuration is suboptimal for the larger model, so we report only the best result from 3 runs. The results are shown below and are also provided in the revision. It can be observed that sharing $B$ consistently outperforms sharing $A$.
>
> | Method                 | ARC-C            | ARC-E            | BoolQ            | HellaS.          | OBQA             | PIQA             | SIQA             | WinoG.           | Avg.                      |
> |------------------------|------------------|------------------|------------------|------------------|------------------|------------------|------------------|------------------|---------------------------|
> | HydraLoRA (Qwen-7B)    | 84.60 $\pm$ 0.30 | 93.20 $\pm$ 0.03 | 73.37 $\pm$ 1.42 | 94.42 $\pm$ 0.36 | 88.30 $\pm$ 0.42 | 89.58 $\pm$ 0.72 | 80.97 $\pm$ 0.06 | 84.33 $\pm$ 0.95 | 86.09 $\pm$ 0.42          |
> | ALoRA (Qwen-7B)        | 85.28 $\pm$ 0.42 | 93.69 $\pm$ 0.30 | 73.41 $\pm$ 0.15 | 94.76 $\pm$ 0.36 | 90.10 $\pm$ 0.42 | 89.07 $\pm$ 0.07 | 80.94 $\pm$ 0.47 | 84.50 $\pm$ 0.50 | **86.47** $\pm$ 0.07 |
> | HydraLoRA (LLaMA2-13B) | 59.90            | 66.46            | 72.42            | 56.46            | 66.80            | 83.95            | 74.26            | 83.50            | 70.47                     |
> | ALoRA (LLaMA2-13B)     | 74.40            | 86.03            | 74.43            | 67.62            | 82.80            | 79.87            | 80.71            | 76.96            | **77.86**           |
>
>
> [1] Q. Dong et al., "MASA: Rethinking the Representational Bottleneck in LoRA with Multi-A Shared Adaptation," arXiv preprint arXiv:2510.06005, 2025.
>
> [2] S. Dou et al., "LoRAMoE: Alleviating world knowledge forgetting in large language models via MoE-style plugin," in ACL, 2024.
>
> [3] Y. Yang et al,, "MTL-LoRA: Low-rank adaptation for multi-task learning," in AAAI, 2025.
>
> [4] Y. Zhou et al., "Cola: Collaborative low-rank adaptation," arXiv preprint arXiv:2505.15471, 2025.

---

### Official Review · Reviewer_aLxW · 2025-10-30

**Soundness:** 3
**Presentation:** 3
**Contribution:** 2
**Rating:** 4
**Confidence:** 3

**Summary:**

This paper proposes a novel multi-task fine-tuning and federated learning approach. By rethinking parameter sharing mechanisms, we design the ALoRA and Fed-ALoRA architectures, which outperform existing methods in both performance and communication efficiency. Research indicates that the similarity in A matrices of LoRAs primarily stems from initialization rather than knowledge sharing, while B matrices play a crucial role in knowledge transfer. Building on this, we propose ALoRA and Fed-ALoRA methods that share B matrices for multi-task and federated fine-tuning respectively. Experiments demonstrate that these methods achieve more balanced performance while maintaining or improving accuracy.

**Strengths:**

1.This study investigates a critical issue on multi-task fine-tuning. The authors investigate the distribution of LoRA parameters in multiple dimensions to support their claim on the contribution of matrices A and B. Therefore, it makes the proposed method well-motivated.
2.The authors have conducted extensive experiments to demonstrate the strong performance of ALoRA and FedA-LoRA compared to baseline methods.

**Weaknesses:**

1. The authors claim that Fed-ALoRA reduces communication costs by 50% and 75% in homogeneous and heterogeneous settings respectively, but the experimental part does not provide corresponding evaluation.
2.The experimental model architecture and scale are limited, lacking exploration of MoE architectures or larger model parameters.

**Questions:**

1. Is it possible for the authors to provide the computational and communication evaluation of Fed-ALoRA？
2. I can hardly derive any real-world application of the proposed method. Pleas provide some cases.

---

> ### Author Response · Authors · 2025-11-28
> **Response to Reviewer aLxW**
>
> We sincerely appreciate your reply. We are committed to addressing your concerns.
>
> **W1 & Q1. The authors claim that Fed-ALoRA reduces communication costs by 50% and 75% in homogeneous and heterogeneous settings respectively, but the experimental part does not provide corresponding evaluation.**
>
> **A**. We compare the number of communicated parameters between Fed-ALoRA and vanilla full-LoRA aggregation in Lines 337–341. In addition, we report the average number of parameters communicated per client in each round in Table 4 and Table 5, based on the LLaMA2-7B model used in our experiments.
>
> **W2. The experimental model architecture and scale are limited, lacking exploration of MoE architectures or larger model parameters.**
>
> **A**. We conduct additional experiments using Qwen2-7B and LLaMA2-13B on commonsense reasoning. We fine-tune the models using the same configuration, and report the mean and standard error over 3 independent runs for Qwen-7B. The results on LLaMA2-13B are less stable, likely because the configuration is suboptimal for the larger model, so we report only the best result from 3 runs. The results are shown below and are also provided in the revision. It can be observed that sharing $B$ consistently outperforms sharing $A$. We adopt the same MoE design as HydraLoRA, since the MoE architecture is not the main focus of our work.
>
> | Method                 | ARC-C            | ARC-E            | BoolQ            | HellaS.          | OBQA             | PIQA             | SIQA             | WinoG.           | Avg.                      |
> |------------------------|------------------|------------------|------------------|------------------|------------------|------------------|------------------|------------------|---------------------------|
> | HydraLoRA (Qwen-7B)    | 84.60 $\pm$ 0.30 | 93.20 $\pm$ 0.03 | 73.37 $\pm$ 1.42 | 94.42 $\pm$ 0.36 | 88.30 $\pm$ 0.42 | 89.58 $\pm$ 0.72 | 80.97 $\pm$ 0.06 | 84.33 $\pm$ 0.95 | 86.09 $\pm$ 0.42          |
> | ALoRA (Qwen-7B)        | 85.28 $\pm$ 0.42 | 93.69 $\pm$ 0.30 | 73.41 $\pm$ 0.15 | 94.76 $\pm$ 0.36 | 90.10 $\pm$ 0.42 | 89.07 $\pm$ 0.07 | 80.94 $\pm$ 0.47 | 84.50 $\pm$ 0.50 | **86.47** $\pm$ 0.07 |
> | HydraLoRA (LLaMA2-13B) | 59.90            | 66.46            | 72.42            | 56.46            | 66.80            | 83.95            | 74.26            | 83.50            | 70.47                     |
> | ALoRA (LLaMA2-13B)     | 74.40            | 86.03            | 74.43            | 67.62            | 82.80            | 79.87            | 80.71            | 76.96            | **77.86**            |
>
> **Q2. I can hardly derive any real-world application of the proposed method. Pleas provide some cases.**
>
> **A**. There are many real-world applications in multi-task and federated fine-tuning. Our study shows that sharing $B$ improves parameter efficiency while preserving empirical performance.

---

### Official Review · Reviewer_i7f4 · 2025-11-01

**Soundness:** 3
**Presentation:** 3
**Contribution:** 2
**Rating:** 4
**Confidence:** 4

**Summary:**

The paper revisits the common belief that, in multi-LoRA setups, A matrices are naturally similar and thus good to share across tasks/clients. Through controlled experiments, the authors show this similarity mainly arises from shared initialization, not shared knowledge; instead, B matrices bear most task-specific information and undergo meaningful learning.
Experiments on multi-task and federated NLP benchmarks show more balanced task performance, competitive or better accuracy, and reduced communication.

**Strengths:**

- Sharing B instead of A is intuitive once the empirical insight is established.
- Applies to both multi-task and federated settings.
- Fed-ALoRA substantially reduces communication (up to 75%) while competitive in performance.

**Weaknesses:**

- Theoretical analysis of why B encodes knowledge while A acts as a projector would help.
- It is unclear which tasks benefit more from A-sharing versus B-sharing.
- The motivation for sharing B is less compelling, especially since the majority of prior literature has focused on sharing A.
- More experiments directly comparing sharing A vs. sharing B are needed to demonstrate the advantage of sharing B.
- Many design choices (e.g., routing strategy, # of A matrices) could benefit from deeper ablation.

**Questions:**

- How sensitive is ALoRA to A/B rank choice-s?

---

> ### Author Response · Authors · 2025-11-28
> **Response to Reviewer i7f4**
>
> We sincerely appreciate your reply. We are committed to addressing your concerns.
>
> **W1. Theoretical analysis of why B encodes knowledge while A acts as a projector would help.**
>
> **A**. Our work focuses on the empirical comparison of how $A$ and $B$ encode knowledge. A related paper [1] studies the expressivity of LoRA. It shows that during fine-tuning, the $A$ matrix is tied to the pre-trained weights, while the $B$ matrix captures the difference between the pre-trained and target weights. This indicates that $B$ carries task-specific information, while $A$ remains relatively agnostic, which is consistent with our empirical findings.
>
> **W2. It is unclear which tasks benefit more from A-sharing versus B-sharing.**
>
> **A**. From our empirical findings, no specific task consistently benefits from sharing $A$ or from sharing $B$. However, sharing $B$ yields more balanced performance overall. This suggests that sharing $B$ encourages more effective knowledge transfer across tasks.
>
> **W3. The motivation for sharing B is less compelling, especially since the majority of prior literature has focused on sharing A.**
>
> **A**. Previous works focus on sharing $A$ because they observed high similarity among $A$ matrices across tasks. However, these works used the same initialization for $A$ (same random seed), which misled them into thinking that $A$ captures shared knowledge across tasks or clients.  We do not simply take the opposite approach. Instead, we provide an extensive study showing that sharing $B$ is more effective than sharing $A$, and this finding motivates our choice to share $B$.
>
> **W4. More experiments directly comparing sharing A vs. sharing B are needed to demonstrate the advantage of sharing B.**
>
> **A**. We conduct additional experiments using Qwen2-7B and LLaMA2-13B on commonsense reasoning.  We fine-tune the models using the same configuration, and report the mean and standard error over 3 independent runs for Qwen-7B. The results on LLaMA2-13B are less stable, likely because the configuration is suboptimal for the larger model, so we report only the best result from 3 runs. The results are shown below and are also provided in the revision. It can be observed that sharing $B$ consistently outperforms sharing $A$.
>
> | Method                 | ARC-C            | ARC-E            | BoolQ            | HellaS.          | OBQA             | PIQA             | SIQA             | WinoG.           | Avg.                      |
> |------------------------|------------------|------------------|------------------|------------------|------------------|------------------|------------------|------------------|---------------------------|
> | HydraLoRA (Qwen-7B)    | 84.60 $\pm$ 0.30 | 93.20 $\pm$ 0.03 | 73.37 $\pm$ 1.42 | 94.42 $\pm$ 0.36 | 88.30 $\pm$ 0.42 | 89.58 $\pm$ 0.72 | 80.97 $\pm$ 0.06 | 84.33 $\pm$ 0.95 | 86.09 $\pm$ 0.42          |
> | ALoRA (Qwen-7B)        | 85.28 $\pm$ 0.42 | 93.69 $\pm$ 0.30 | 73.41 $\pm$ 0.15 | 94.76 $\pm$ 0.36 | 90.10 $\pm$ 0.42 | 89.07 $\pm$ 0.07 | 80.94 $\pm$ 0.47 | 84.50 $\pm$ 0.50 | **86.47** $\pm$ 0.07 |
> | HydraLoRA (LLaMA2-13B) | 59.90            | 66.46            | 72.42            | 56.46            | 66.80            | 83.95            | 74.26            | 83.50            | 70.47                     |
> | ALoRA (LLaMA2-13B)     | 74.40            | 86.03            | 74.43            | 67.62            | 82.80            | 79.87            | 80.71            | 76.96            | **77.86**            |
>
>
> **W5 & Q1. Many design choices (e.g., routing strategy, # of A matrices) could benefit from deeper ablation. How sensitive is ALoRA to A/B rank choices?**
>
> **A**. The primary goal is to demonstrate the structural benefits of sharing $B$ instead of $A$, which we have already shown in our paper. Due to the time limits, we will add more experiments in our later revision, including an ablation on the number of $A$ matrices and a sensitivity study on the rank choices.
>
>
> [1] Zeng, Yuchen, and Kangwook Lee. "The expressive power of low-rank adaptation." arXiv preprint arXiv:2310.17513 (2023).

---

### Official Review · Reviewer_CEiW · 2025-11-01

**Soundness:** 1
**Presentation:** 2
**Contribution:** 2
**Rating:** 2
**Confidence:** 3

**Summary:**

This paper re-examines the common practice of sharing the LoRA `A` matrix in multi-adapter settings. The authors hypothesize that the observed similarity in `A` matrices is an artifact of identical initialization, not shared knowledge, and argue the `B` matrix is more critical for knowledge encoding. Based on this, they propose ALoRA, an asymmetric architecture with multiple task-specific `A` matrices and a single shared `B` matrix. For federated learning, they introduce Fed-ALoRA, which communicates only `B` matrices, and includes a matrix decomposition strategy to handle heterogeneous LoRA ranks. Experiments on multi-task and federated benchmarks are presented to support the claim that these methods achieve more balanced performance with comparable or better average accuracy.

**Strengths:**

*   **Provocative Re-evaluation of LoRA Dynamics:** The paper's primary merit is its willingness to question a common assumption in multi-LoRA methods. The initial analysis in Section 3.1, which investigates the differing stability of `A` and `B` matrices under different initializations (Figure 1), poses an interesting and provocative question that challenges the rationale behind `A`-sharing architectures like HydraLoRA.

*   **Practical Design for Heterogeneous Federated Learning:** The proposed Fed-ALoRA method considers a critical real-world problem: enabling collaboration among clients with heterogeneous LoRA ranks. The use of matrix decomposition (Sec 4.2) to standardize the shared parameter space demonstrates thoughtful engineering to address a practical limitation of prior works.

**Weaknesses:**

*   **Fundamental Contradiction Between Motivation and Method:** The paper's core logic is self-contradictory. The authors' own analysis (Sec 3.2, Figure 2) posits that the `A` matrix is largely static ("fixed feature projector") while the `B` matrix is dynamic and encodes task-specific knowledge. The most logical conclusion from this finding would be to share the common, static `A` matrix across tasks/clients while allowing the dynamic `B` matrix to be learned independently. Instead, the paper proposes the exact opposite—sharing `B` and keeping `A` separate—without providing a convincing theoretical justification for this inversion. This strongly suggests that the method was not derived from the paper's stated motivation, but rather that the motivation was retrofitted to explain empirical results, which undermines the scientific integrity of the work.

*   **Incremental Contribution and Overstated Novelty:** The core idea of Fed-ALoRA is far from novel and appears to be a derivative of existing work. Architectures that selectively share a single LoRA matrix to reduce communication costs are already established, most notably **FedSA-LoRA** (which shares `A`). The proposed method is essentially a "symmetric inversion" of this concept. Furthermore, the technique used to handle heterogeneity—matrix decomposition to align dimensions—is a standard and widely used tool in machine learning for dealing with mismatched parameter spaces. The paper fails to demonstrate a significant conceptual leap, positioning its contribution as incremental at best.

*   **Fundamentally Flawed and Irrelevant Baseline Comparisons:** The experimental design is compromised by unfair and irrelevant comparisons that create a misleading narrative.
    1.  **Orthogonal Problem Comparison:** The inclusion of **AdaLoRA** as a baseline is highly questionable. AdaLoRA addresses the problem of *intra-task* dynamic rank allocation for parameter efficiency. This paper addresses *inter-task/client* parameter sharing for knowledge transfer and communication efficiency. These are orthogonal research problems. Comparing ALoRA to AdaLoRA does not isolate the benefits of `B`-sharing; it merely shows that a parameter-sharing method can outperform a sophisticated but non-sharing method, which is an expected but uninformative result. This suggests a "kitchen-sink" approach to baseline selection rather than principled experimental design.
    2.  **Apples-to-Oranges Comparison:** The ALoRA vs. HydraLoRA comparison (Table 2) is misleading. ALoRA is an MoE-like architecture with a routing mechanism, whereas HydraLoRA is a simpler parameter-sharing model. Attributing performance differences solely to the choice of shared matrix (`B` vs. `A`) is disingenuous when the architectural complexities are fundamentally different.
    3.  **Strawman Argument:** The comparison in the heterogeneous setting (Table 5) is built upon a strawman. The authors admit the baseline `FedSA-LoRA*` is their own adaptation of a method not designed for heterogeneity. Its poor performance cannot be reliably interpreted as a failure of `A`-sharing.

*   **Lack of Statistical Rigor Renders Results Unreliable:** The paper reports single-run point estimates for all experiments (Tables 2, 3, 4, 5). This is a critical methodological flaw. Given the high variance inherent in LLM fine-tuning, the small margins of improvement reported are statistically meaningless without mean scores and standard deviations over multiple runs. It is impossible to verify whether the claimed gains are real or simply artifacts of random chance.

*   **Foundational Claims Rest on Extremely Narrow Evidence:** The central claims about the universal roles of matrices `A` and `B` are based on an extremely limited experimental setup (LLaMA2-7B on the Dolly-15K dataset). These findings are presented as general properties of LoRA, but there is no evidence to support their generalizability across recent open source models, different model architectures, model sizes, or a wider variety of task domains.

*   **Ambiguity in the Heterogeneous Federated Learning Method:** The formulation and update procedure for the heterogeneous Fed-ALoRA method are not sufficiently clear.
    1.  The role of the accumulator $B_i^0$ is confusing. The local training objective is given as $\mathcal{L}(W_0 + (B_i^0 + B_{i2}B_{i1})M_iA_i)$ (Sec 4.2, Step 2). However, Step 4 states the server broadcasts a global matrix $B_0^t$. It is not specified how client $i$ uses this $B_0^t$ to update its local accumulator $B_i^0$ for the next round. Does $B_i^0$ become $B_0^t$, or is it accumulated (e.g., $B_i^0 \leftarrow B_i^0 + B_0^t$)? The text only says $B_i^0$ is initialized with $B_{t-1}^0$ (Step 1), which seems to imply it never receives the global update.
    2.  The paper does not provide a clear justification for the specific decomposition $\Delta W_i = B_{i2}B_{i1}M_iA_i$. While it enables aggregation, the introduction of three new matrices ($B_{i1}$, $B_{i2}$, $M_i$) increases complexity. The rationale for this particular design over other potential solutions (e.g., projecting heterogeneous $B$ matrices to a common subspace) is not discussed.

**Questions:**

1.  Your own analysis suggests the `A` matrix is static while `B` is dynamic. The intuitive conclusion would be to share the static part (`A`). Your method does the opposite. Could you resolve this apparent **logical contradiction**? Why is it better to share the dynamic, knowledge-encoding part across different clients/tasks?

2.  Could you clarify the rationale for including **AdaLoRA** as a baseline? AdaLoRA solves the orthogonal problem of dynamic rank allocation within a single task, while your work focuses on parameter sharing across tasks. What specific hypothesis is this comparison intended to test, other than showing that parameter sharing can outperform no sharing?

3.  Given prior work like **FedSA-LoRA** (which shares `A`), your proposed Fed-ALoRA can be seen as a "reverse design." Could you clarify the novel scientific contribution of your work beyond this inversion and the application of standard matrix decomposition techniques for handling heterogeneity?

4.  In the heterogeneous setting (Table 5), you compare against `FedSA-LoRA*`, which is your own adaptation. Do you agree that this is not a fair or valid comparison? How can you justify claims of superiority when the baseline's poor performance might simply be an artifact of a suboptimal adaptation?

5.  All reported results lack error bars. Given that the reported performance gains are often marginal, how can you be confident that these results are **statistically significant** and not just noise from a single random seed?

---

> ### Author Response · Authors · 2025-11-28
> **Response to Reviewer CEiW**
>
> We sincerely appreciate your reply. We are committed to addressing your concerns.
>
>
>  **W1: Fundamental Contradiction Between Motivation and Method**
>
> **A**: We observed that $A$ matrix remain largely static, while the $B$ matrix is dynamic and encodes task-specific knowledge. This finding contrasts with the previously common belief that $A$ captures task knowledge, which motivated the widely used sharing-$A$ design. Our results suggest instead that $A$ acts as a fixed feature projector, whereas $B$ aggregates the features and thus is more informative. Therefore, we are wondering whether $A$ or $B$ should be shared. The sharing-$A$ design may limit the model's ability to explore diverse feature subspaces. That's our motivation, and we believe the logic is consistent.
>
> **W2: Incremental Contribution and Overstated Novelty**
>
> **A**: As our paper highlights, we provide a novel perspective by revisiting which parameter in LoRA plays a more dominant role in effective knowledge sharing. This aspect has been overlooked in previous work. The proposed Fed-ALoRA can be viewed as a symmetric inversion of FedSA-LoRA in implementation. But it is motivated by our finding that sharing the $B$ matrix improves knowledge sharing. Moreover, FedSA-LoRA supports only the homogeneous setting, while our Fed-ALoRA supports both homogeneous and heterogeneous settings. To the best of our knowledge, our matrix decomposition strategy is novel in handling heterogeneous problems in this way, and there is little prior work using similar approaches.
>
> **W3: Fundamentally Flawed and Irrelevant Baseline Comparisons**
>  - **W3.1 Orthogonal Problem Comparison**
>  - **W3.2 The ALoRA vs. HydraLoRA comparison (Table 2) is misleading. ALoRA is an MoE-like architecture with a routing mechanism, whereas HydraLoRA is a simpler parameter-sharing model. Attributing performance differences solely to the choice of shared matrix (B vs. A) is disingenuous when the architectural complexities are fundamentally different.**
>  - **W3.3 Strawman Argument: The comparison in the heterogeneous setting (Table 5) is built upon a strawman. The authors admit the baseline FedSA-LoRA\* is their own adaptation of a method not designed for heterogeneity. Its poor performance cannot be reliably interpreted as a failure of A-sharing.**
>
> **A**:
>  - We include AdaLoRA because it is a competitive and commonly used baseline. The comparison between ALoRA and AdaLoRA shows the benefit of parameter sharing. In addition, the comparison between ALoRA and HydraLoRA shows the structural benefit of sharing the $B$ matrix.
>  - Both ALoRA and HydraLoRA use the same MoE structure. The only difference is whether $A$ or $B$ matrix is shared. Both methods have the same architectural complexity.
>  - FedSA-LoRA does not support the heterogeneous setting, while we introduce a novel matrix decomposition strategy that enables it. To compare FedSA-LoRA (sharing $A$) and Fed-ALoRA (sharing $B$) under this setting, we apply the same strategy to FedSA-LoRA. There are no other differences; all experiments use the same configuration.

---

> ### Author Response · Authors · 2025-11-28
> **Response to Reviewer CEiW (cont.)**
>
> **W4: Lack of Statistical Rigor Renders Results Unreliable**
>
> **A**: We reran the experiments in Table 2 two more times for each method using different random seeds. We then report the mean and standard error, shown below. ALoRA achieves the most balanced performance. Due to time limits, we will complete the statistical error information for the remaining tables in our later revision.
>
> | Method       | ARC-C            | ARC-E            | BoolQ            | HellaS.          | OBQA             | PIQA             | SIQA             | WinoG.           | Avg.                      | $\Delta m\%(\downarrow)$ |
> |--------------|------------------|------------------|------------------|------------------|------------------|------------------|------------------|------------------|---------------------------|--------------------------|
> | Single       | 77.76 $\pm$ 0.97 | 90.81 $\pm$ 0.23 | 73.39 $\pm$ 1.05 | 95.40 $\pm$ 0.08 | 86.60 $\pm$ 0.72 | 89.25 $\pm$ 0.47 | 80.69 $\pm$ 0.48 | 85.35 $\pm$ 0.64 | 84.90 $\pm$ 0.21          |                          |
> | LoRA         | 76.66 $\pm$ 1.02 | 88.72 $\pm$ 0.83 | 72.73 $\pm$ 1.24 | 94.50 $\pm$ 0.41 | 84.10 $\pm$ 0.14 | 87.59 $\pm$ 0.15 | 79.42 $\pm$ 0.15 | 85.40 $\pm$ 1.00 | 83.64 $\pm$ 0.27          | 1.48                     |
> | LoHa         | 77.13 $\pm$ 0.13 | 88.91 $\pm$ 0.09 | 72.89 $\pm$ 0.19 | 94.05 $\pm$ 0.23 | 85.40 $\pm$ 0.56 | 87.84 $\pm$ 0.34 | 78.92 $\pm$ 0.44 | 84.73 $\pm$ 0.17 | 83.73 $\pm$ 0.11          | 1.36                     |
> | AdaLoRA      | 77.72 $\pm$ 0.72 | 89.72 $\pm$ 0.96 | 73.32 $\pm$ 0.19 | 93.98 $\pm$ 0.73 | 85.20 $\pm$ 1.13 | 87.90 $\pm$ 0.35 | 79.45 $\pm$ 0.18 | 83.78 $\pm$ 0.28 | 83.88 $\pm$ 0.32          | 1.17                     |
> | MoSLoRA      | 76.87 $\pm$ 0.12 | 89.22 $\pm$ 0.15 | 73.50 $\pm$ 0.86 | 95.02 $\pm$ 0.13 | 85.27 $\pm$ 2.58 | 87.99 $\pm$ 0.59 | 80.89 $\pm$ 0.12 | 85.13 $\pm$ 0.55 | 84.23 $\pm$ 0.39          | 0.76                     |
> | HydraLoRA    | 78.58 $\pm$ 0.12 | 89.94 $\pm$ 0.18 | 75.02 $\pm$ 0.18 | 95.21 $\pm$ 0.11 | 84.90 $\pm$ 1.55 | 88.03 $\pm$ 0.15 | 79.99 $\pm$ 0.72 | 84.92 $\pm$ 0.34 | 84.57 $\pm$ 0.17          | 0.32                     |
> | ALoRA (ours) | 79.40 $\pm$ 0.41 | 89.69 $\pm$ 0.71 | 74.38 $\pm$ 0.10 | 94.85 $\pm$ 0.23 | 86.10 $\pm$ 0.14 | 88.24 $\pm$ 0.49 | 80.17 $\pm$ 0.32 | 85.68 $\pm$ 0.27 | **84.81** $\pm$ 0.03 | **0.04**
>
> **W5: Foundational Claims Rest on Extremely Narrow Evidence**
>
> **A**: We provide the training dynamics of A and B using the Qwen-7B model [1] on the bigscience/xp3 dataset [2], which contains data from 46 languages and 16 NLP tasks. We sample 3,000 English examples and fine-tune the Qwen-7B model for 2 epochs.
>
> Following Section 3.2, we analyze the magnitude and direction changes during training. The results are provided in the revision. We observe the same pattern: the B matrix plays a more dominant role than the A matrix during training.
>
> **W6: Ambiguity in the Heterogeneous Federated Learning Method. 1. The role of the accumulator $B_i^0$ is confusing. 2. The paper does not provide a clear justification for the specific decomposition $\Delta W_i=B_{i2}B_{i1}M_iA_i$.**
>
> **A*: *
>  - The accumulator $B_{i0}$ stores all global updates received so far. This means that in each round, it is updated as $B_{i0}\leftarrow B_{i0} + B_i^t$.
>  - The decomposition strategy is simple but effective, as it enables heterogeneous federated learning efficiently. It introduces $2 d_m r$ additional parameters, but since $d_m$ is chosen to be comparable to $r$, the increase in complexity is minimal. In contrast, subspace-based methods require subspace computation and projection, which adds more computational cost than our approach.
>
> [1] Yang, An, et al. "Qwen3 technical report." arXiv preprint arXiv:2505.09388 (2025).
>
> [2] Muennighoff, Niklas, et al. "Crosslingual generalization through multitask finetuning." Proceedings of the 61st Annual Meeting of the Association for Computational Linguistics (Volume 1: Long Papers). 2023.

---

### Note · Authors · 2026-01-03

**Comment:**

Thanks for the valuable feedback from reviewers and AC. We have decided to withdraw our submission.

Best regards,
Authors

**Withdrawal Confirmation:**

I have read and agree with the venue's withdrawal policy on behalf of myself and my co-authors.